# Averaging Is Probably Not the Optimum Way of Aggregating Parameters in Federated Learning

**DOI:** 10.3390/e22030314

**Published:** 2020-03-11

**Authors:** Peng Xiao, Samuel Cheng, Vladimir Stankovic, Dejan Vukobratovic

**Affiliations:** 1Department of Computer Science and Technology, Tongji University, Shanghai 201804, China; phd.xiaopeng@gmail.com; 2The School of Electrical and Computer Engineering, University of Oklahoma, Tulsa, OK 73019, USA; 3Department of Electronic and Electrical engineering, University of Strathclyde, Glasgow G1 1XW, UK; vladimir.stankovic@strath.ac.uk; 4Faculty of Technical Sciences, University of Novi Sad, 21000 Novi Sad, Serbia; dejanv@uns.ac.rs

**Keywords:** federated learning, decentralized learning, averaging, mutual information, correlation

## Abstract

Federated learning is a decentralized topology of deep learning, that trains a shared model through data distributed among each client (like mobile phones, wearable devices), in order to ensure data privacy by avoiding raw data exposed in data center (server). After each client computes a new model parameter by stochastic gradient descent (SGD) based on their own local data, these locally-computed parameters will be aggregated to generate an updated global model. Many current state-of-the-art studies aggregate different client-computed parameters by averaging them, but none theoretically explains why averaging parameters is a good approach. In this paper, we treat each client computed parameter as a random vector because of the stochastic properties of SGD, and estimate mutual information between two client computed parameters at different training phases using two methods in two learning tasks. The results confirm the correlation between different clients and show an increasing trend of mutual information with training iteration. However, when we further compute the distance between client computed parameters, we find that parameters are getting more correlated while not getting closer. This phenomenon suggests that averaging parameters may not be the optimum way of aggregating trained parameters.

## 1. Introduction

Nowadays, more and more intelligent devices, such as smart phones, wearable devices and autonomous vehicles, are widely used [1,2], generating a wealth of data. The generated data can be used to develop deep learning models powering applications such as speech recognition, face detection and text entry. With the increasing amount of generated data and increasing computing power of the smart devices [3], recent studies have explored distributed training of models at these edging devices [4,5]. Considering that centralized storage of data in the central server, e.g., in the cloud, is often not feasible due to privacy of consumer data [6], *federated learning* [7] has been advocated as an alternative setting.

Federated learning [7] can be viewed as an extension of conventional distributed deep learning [8], as it aims to train a high quality shared model while keeping data distributed over clients. That is, each client computes an updated model based on his/her own locally collected data (that is not shared with others). After all model parameters are computed locally, they are sent to the server where an updated global model is computed by combining the received local model parameters.

Federated learning plays a critical role in various privacy-sensitive scenarios, such as power intelligence applications like keyboard prediction and emoji prediction in smart phone [9,10], optimising patient’s healthcare in hospitals [11], helping internet of things (IoT) systems adapt to environmental changes [12] etc. Specifically, assume there are *Q* participating clients over which the data is partitioned, with $P_{q}$ being the set of indices of data points at client *q*, and $n_{q}=\left|P_{q}\right|$. A federated learning system optimizes a global model summarized as follows:(1)All participating clients download the latest global model parameter vector $w^{,}$.(2)Each client improves downloaded model based on their local data using, e.g., stochastic gradient descent (SGD) [13] with a fixed learning rate η: $w_{q}=w^{,}-\eta g_{q}$, where $g_{q}=▽F_{q}\left(w^{,}\right)$, and $F_{q}\left(w\right)$ is the local objective function for the *q*th device.(3)Each improved model parameter $w_{q}$ is sent from the client to the server.(4)The server aggregates all updated parameters to construct an enhanced global model $w^{*}$.

Current studies on federated learning and its variants address a series of challenges. In [14], they compress the model in both upstream and downstream links to promote communication efficiency. In [15], they demonstrate user-level differential privacy in large recurrent language models. In [16], they propose algorithms that achieve both communication efficiency and differential privacy. In [17], they propose a decentralized training framework to overcome issues related to heterogeneous devices. In [18], they suggest a general oracle-based framework in parallel stochastic optimization. In [19], they investigate a pluralistic solution to address the cyclic patterns in data samples during federated training. In these state-of-the-art studies, at step 4) above, the updated model parameters are calculated by directly averaging the models received from all clients via $w^{*}=\sum_{q=1}^{\left|Q\right|}\frac{n_{q}}{n}w_{q}$, where $n=\sum_{q=1}^{\left|Q\right|}n_{q}$. However, after each client computes $w_{q}$ by optimizing their local objective function $F_{q}$, there is no evidence $w^{*}$ will be close to the optimized value of the global objective function $\sum_{q}\frac{n_{q}}{n}F_{q}\left(w\right)$. That raises a question: is averaging model parameters a reasonable approach in federated learning? With this question we begin our research.

Mutual information (MI) is a powerful statistic for measuring the degree of relatedness [20]. MI can detect any kind of relationship between random variables even those that do not manifest themselves in the covariance [21], and has a straightforward interpretation as the amount of shared information between datasets (measured in, for example, bits) [22]. In this paper, we treat each client’s computed parameter vector $w_{q}$ as an random vector because of the stochastic properties of SGD, and train each client in one training epoch many times to obtain a series of parameter samples. We firstly calculate the MI between different parameters in two learning tasks: (1) convolutional neural network (CNN) [23] designed for a classification task and (2) recurrent neural network (RNN) [24] on a regression task. By using two methods, one is base on the MI derivative formula under the multi-dimensional Gaussian distribution assumption, and another is a continuous-continuous MI estimator based on k-nearest neighbor (KNN) distances described in [21]. We estimate MI at different training phases. The results confirm the correlation between the model parameters at different clients, and show an increasing trend of MI. We further calculate the distance variation between different parameters using three regular distance metrics: Euclidean distance [25], Manhattan distance [26], Chebyshev distance [27]. By comparing the distance variation with MI variation, we show that the parameters are getting more correlated while not getting closer. A proposition derived in Section 3 implies that averaging parameters is not supported by theory and requires further scrutiny.

The rest of the paper is organized as follows. In Section 2, we provide an overview of related work. In Section 3, we describe the details of estimating MI. Section 4 shows the experiment results. Conclusion and future work are discussed in Section 5.

## 2. Related Work

In [28], the authors represent Deep Neural Networks (DNNs) as a Markov chain and propose to use information theoretical tools, and in particular, Information Plane (IP)–the plane of the MI values of any other variable with the input variable (input data) and desired output variable (label), to study and visualise DNNs. In [29], they extend the approach of [28] demonstrating the effectiveness of the Information Plane visualization of DNNs. In particular, the authors of [29] suggest that the training process of DNNs is composed of an initial fitting phase and a representation compression phase, where the goal of the network is to optimize the Information Bottleneck (IB) tradeoff between compression and prediction. These results are later questioned by [30], who argue that the compression phase is not general in the DNN training process. However, [31] propose ensemble dependency graph estimator (EDGE) to estimate MI supporting the claim of the compression phase. Following this line of work, [32] propose a method for performing IP on arbitrarily-distributed discrete and/or continuous variables, while [33] use matrix-based $R\overset{´}{e}nyi’s$ entropy to provide comprehensive IP analysis of large DNNs.

There are few other attempts to study federated learning from an information-theoretic perspective. In [34], they apply information theory to understand the aggregation process of DNNs in the federated learning setup. The authors use EDGE [31] to measure the MI between representation and inputs as well as representation and labels in each local model, and compare them to the respective information contained in the representation of the averaged model. In [34], they leave as an open question for the theoretical analysis of when and how to aggregate local models. In contrast to [34] and the other work reviewed above, in this paper, we treat each local model as a continuous random vector, and directly measure the MI between different local models through the training phase using two methods. By doing so we explore the correlation between each local model. By further comparing with the distance variation between local models, we propose that averaging may not be the optimum way of aggregating trained parameters.

## 3. Estimating the Mutual Information

In this section, we show two methods to estimate MI between model parameters at two clients. To do that, we need to perform training at each client many times to acquire enough samples of each parameter vector. Generally speaking, let *X* and *Y* be the model parameters computed at two different clients, $X,Y\in R^{t}$. Let $Z=\left(X,Y\right),Z\in R^{2t}$ be the joint variable of *X* and *Y*. After training at the two clients *N* times, we get *N* samples of *X* and *Y*, denoted by $x_{1},x_{2},\ldots ,x_{N}$ and $y_{1},y_{2},\ldots ,y_{N}$, and also *N* samples of *Z*, $z_{1},z_{2},\ldots ,z_{N},z_{i}=\left(x_{i},y_{i}\right)$.

### 3.1. MI Derivative Formula under Multi-D Gaussian

In the first case, we assume that both *X* and *Y* satisfy the multi-dimensional Gaussian distribution, i.e., $X∼N(\mu_{x},\Sigma_{x})$, $Y∼N(\mu_{y},\Sigma_{y})$, which implies that *Z* also satisfies the multi-dimensional Gaussian distribution $Z∼N(\mu_{z},\Sigma_{z})$.

Firstly, we calculate *Z*’s covariance matrix $\Sigma_{z}$ according to the *N* samples: (1)$$\Sigma_{z}=\left(\begin{array}{cccccc} cov(X^{1},X^{1}) & \cdots & cov(X^{1},X^{t}) & cov(X^{1},Y^{1}) & \cdots & cov(X^{1},Y^{t})\\ \vdots & \ddots & \vdots & \vdots & \ddots & \vdots \\ cov(X^{t},X^{1}) & \cdots & cov(X^{t},X^{t}) & cov(X^{t},Y^{1}) & \cdots & cov(X^{t},Y^{t})\\ cov\left(Y^{1},X^{1}\right) & \cdots & cov\left(Y^{1},X^{t}\right) & cov\left(Y^{1},Y^{1}\right) & \cdots & cov\left(Y^{1},Y^{t}\right)\\ \vdots & \ddots & \vdots & \vdots & \ddots & \vdots \\ cov\left(Y^{t},X^{1}\right) & \cdots & cov\left(Y^{t},X^{t}\right) & cov\left(Y^{t},Y^{1}\right) & \cdots & cov\left(Y^{t},Y^{t}\right) \end{array}\right),$$ where $X^{i},Y^{j},i,j=1,2,\ldots ,t$ denote *i*th and *j*th dimension of *X* and *Y*, and $cov\left(X^{i},Y^{j}\right)=E\left[X^{i}-E\left[X^{i}\right]\right]\left[Y^{j}-E\left[Y^{j}\right]\right]$ denotes the covariance between $X^{i}$ and $Y^{j}$. The mean of each variable is estimated from the collected samples.

Then, to calculate H(Z)=H(X,Y), we use the entropy formula for multi-dimensional Gaussian distribution [35]:(2)$$H\left(Z\right)=E\left[-\log p\left(Z\right)\right]=\log\sqrt{det(2\pi e\Sigma_{z})},Z∼N\left(\mu_{z},\Sigma_{z}\right).$$Since *Z* is the joint distribution of *X* and *Y*, from Equation (1) we can infer that $\Sigma_{x}=\Sigma_{z}\left[1:t,1:t\right],\Sigma_{y}=\Sigma_{z}\left[t+1:2t,t+1:2t\right]$, and through Equation (2), we then calculate H(X) and H(Y) as:(3)$$\begin{array}{c} H\left(X\right)=\log\sqrt{det(2\pi e\Sigma_{z}\left[1:t,1:t\right])}, \end{array}$$(4)$$\begin{array}{c} H\left(Y\right)=\log\sqrt{det(2\pi e\Sigma_{z}\left[t+1:2t,t+1:2t\right])}. \end{array}$$

Finally, through the relationship formula between entropy and MI [35],
(5)I(X,Y)=H(X)+H(Y)-H(X,Y),
we calculate the mutual information between *X* and *Y*.

### 3.2. KNN Discretization Estimator

In the previous subsection, we estimated MI under multi-Gaussian assumption. In this subsection, we show how to estimate MI with unknown distribution by the averaging *k*-nearest neighbor distance [21], which is one of the most commonly used continuous-continuous MI discretization estimators. Note that there are other methods to estimate MI, such as EDGE [31]; however, EDGE method [31] is better for estimating a continuous-discrete mixture MI.

Specifically, let us denote by ϵ(i) the distance from $z_{i}$ to its *k*th neighbor. We count the number $n_{x}\left(i\right)$ of points, $x_{j}$, whose distance to $x_{i}$ is strictly less than ϵ(i)/2, and similarly for *Y*. The estimate for MI is then:(6)$$I\left(X,Y\right)=\psi \left(k\right)-\frac{1}{N}\sum_{i=1}^{N}\left(\psi \left(n_{x}\left(i\right)+1\right)+\psi \left(n_{y}\left(i\right)+1\right)\right)+\psi \left(N\right),$$
where ψ(·) is the digamma function, and *k* is a hyperparameter. In this paper we choose *k* to 3 after tuning. This method is mainly using the probability of points being in the *k*th nearest distance, to approximately approach the true probability density, and has an advantage in vastly reducing systematic errors. For more details please refer to [21].

## 4. Simulation Results

In simulations, considering the time it takes to train a model and the limitations of our hardware, we selected two relatively small tasks and perform them in local iid dataset setting. One is training a simple RNN model to predict the shift and scale variation of a signal sequence. The second one is applying transfer learning to train a CNN model on the MNIST [36] digit recognition dataset. For both tasks we train model parameters at three clients (indexed by 0, 1, 2) in a whole training epoch many times, and estimate the MI between three model parameters at different training phases.

### 4.1. RNN on Signal Variation Prediction

In the first experiment, we consider training an RNN model to predict the shift and scale variation of an arbitrary randomized 1-d signal sequence. Each client contains 100 training examples, where each training example is a signal sequence of length 64, generated by randomizing the source. The model is initialized by the normal initializer, and optimized by the Adam [37] optimizer with learning rate 0.001. The mini-batch size is 64. The total number of samples obtained for each client is 22,000.

The model architecture we use is a simple RNN with 10 layers and four units, where each layer is wrapped by Dropout function [38], for a total of 394 parameters. The metric we use is the mean squared error (MSE). Though this model is not state of the art, it is sufficient for our purpose, as our goal is to estimate the MI between clients, not to achieve the best possible performance for this task.

Figure 1a,b shows the MI estimations in the regression task by two methods. The unit of MI is bit. Client(0, 1), Client(0,2), Client(1, 2), respectively, indicate MI between model parameters as Client 0 and 1, Client 0 and 2, Client 1 and 2. From the results of both methods, we can see that the MI between all of three clients are always larger than zero throughout the whole training epoch, which shows the correlation between different clients in the federated mode. Figure 1c shows the decrease of MSE with training iteration. All three models were trained well in their own data, and all of them eventually converge to a similar MSE with similar training patterns. Comparing to similar MSE variations, Figure 1a,b show the difference of MI between different client-pairs.

### 4.2. CNN on Digit Recognition

In the second experiment, we firstly train the Lenet model [39] from scratch on the whole MNIST dataset. After getting the pretrained Lenet model, at each client, we keep the trained Lenet model unchanged except the last fully connected layer and apply it on augmented MNIST data. This way, we reduce the number of training parameters to 850. Each client contains 200 training examples (20 for each digit) and each training example is an augmentation of a randomized selection of an original digit image in MNIST. The augmented operation includes rotation, zoom, shift (all of these operations are performed in a certain range). The model is optimized by Adam optimizer with learning rate 1e-5. The mini-batch size is 50. The total number of samples obtained at each client is 26,400. The metric we use is recognition accuracy.

Figure 2a,b shows the obtained MI estimation. The unit of MI is bit. Similar to the previous experiment, the MI between all of three clients is larger than zero throughout the training phase, which shows that there exists correlation between the clients’ parameter models. Figure 2c shows improvement in recognition accuracy with training iteration. All of three models also have similar training pattern like in regression tasks, but still looks different in MI estimations. These imply that although different clients in a same learning task have similar training trajectory, the MI can still the subtle difference between them.

### 4.3. Distance Metrics

The MI results presented in the previous section indicate an upward trend of correlation between different client parameters with the training iterations. The increase in correlation seems to imply that the parameters trained from different clients converge. To verify this, in this subsection, we measure directly the difference of the parameter models at different clients using three different metrics.

We detect the distance between the model parameters at different clients using three regular distance metrics: Euclidean distance, Manhattan distance, Chebyshev distance, and refer to them as $D_{Eu},D_{Man},D_{Che}$, respectively. Specially, to calculate the distance between two model parameters, we firstly calculate the distance between two samples of the client computed parameters, and then average the result over all samples, i.e.,:(7)$$\begin{array}{c} D_{Eu}=\frac{1}{N}\sum_{i=1}^{N}\sqrt{\sum_{k=1}^{t}{\left(x_{i}^{k}-y_{i}^{k}\right)}^{2}}, \end{array}$$(8)$$\begin{array}{c} D_{Man}=\frac{1}{N}\sum_{i=1}^{N}\sum_{k=1}^{t}|x_{i}^{k}-y_{i}^{k}|, \end{array}$$(9)$$\begin{array}{c} D_{Che}=\frac{1}{N}\sum_{i=1}^{N}\max_{k}\left(\right|x_{i}^{k}-y_{i}^{k}\left|\right). \end{array}$$

The results of distance variation for the two learning tasks can be seen in Figure 3 and Figure 4. As we can see, for both learning tasks, all three distance metrics between any client pairs increase rather than decrease with the training iteration. This may not be surprising as we initialize the parameters to be the same across all clients. However, this contradicts with the conclusion of the previous section that the parameters converge. Studying the trend of ML and distance change with the training iteration, the following proposition is derived:

**Proposition** **1.**
*In federated learning, the model parameters of two clients X and Y cannot be modelled by the following additive model: $X=P+N_{1},Y=P+N_{2}$, where $N_{1}$ and $N_{2}$ are independent noises and independent of some optimum set of parameters P.*


**Proof.** Let us assume that *X* and *Y* satisfy the proposed additive model, then $X=Y+N_{1}-N_{2}$. Mutual information between *X* and *Y* increases as noise variances of $N_{1}$ and $N_{2}$ decreases, which also leads to decreasing distance between *X* and *Y*. This contradicts our simulation results shown in Figure 3 and Figure 4. Therefore, the model parameters of the two clients cannot be modelled in the proposed form. □

Averaging is a very restrictive estimate. According to Proposition 1, if the parameters cannot be the modelled using an additive noise model, there is no particular reason why averaging different parameters would result in the optimal or even a reasonably good set of parameters. Although averaging parameters achieved good results in some learning tasks, it is not a best or even reasonable choice in general federated learning.

## 5. Conclusions

This study explores the correlation between clients’ model parameters in federated learning. Through estimating the MI between different client computed parameters in two learning tasks by two methods, we confirm the existence of correlation between different clients. By further studying variation of three distance metrics, we show that model parameters at different clients are getting more correlated while not getting closer as with training iterations. All of these implies that the aggregating parameters by averaging the local model parameters of each client directly may not be a reasonable approach in general.

We encourage more studies to explore the federated learning through information-theoretic perspective in large learning scenarios and non-iid setting. The existence of correlation implies the possibility that distributed source coding can be applied to compress the information in federated learning. Thus we can explore compressing information to promote communication efficiency in federated learning. The derived Proposition also implies there maybe some better choice than averaging parameters in the federated mode, which can be a future direction in federated learning.

## Figures and Tables

**Figure 1 entropy-22-00314-f001:**
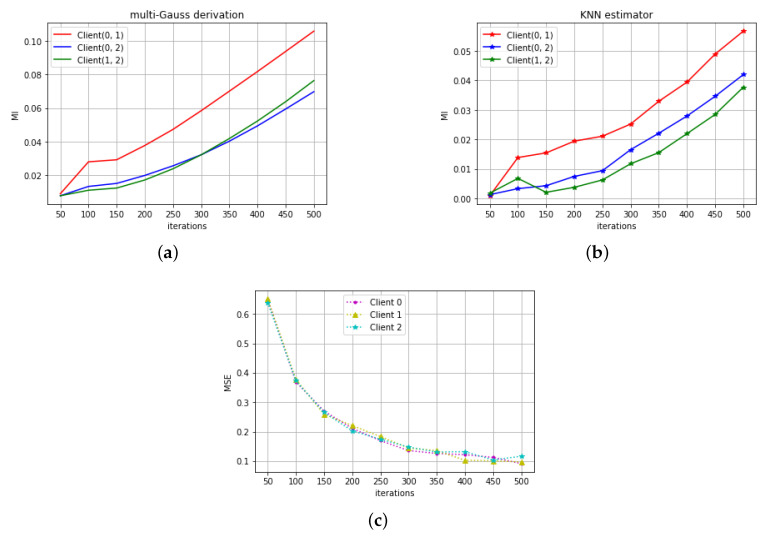
(**a**) The MI in the signal variation prediction task calculated by the first method (Section 3.1). (**b**) The MI in the signal variation prediction task calculated by the second method (Section 3.2). (**c**) The MSE variation with the training iterations.

**Figure 2 entropy-22-00314-f002:**
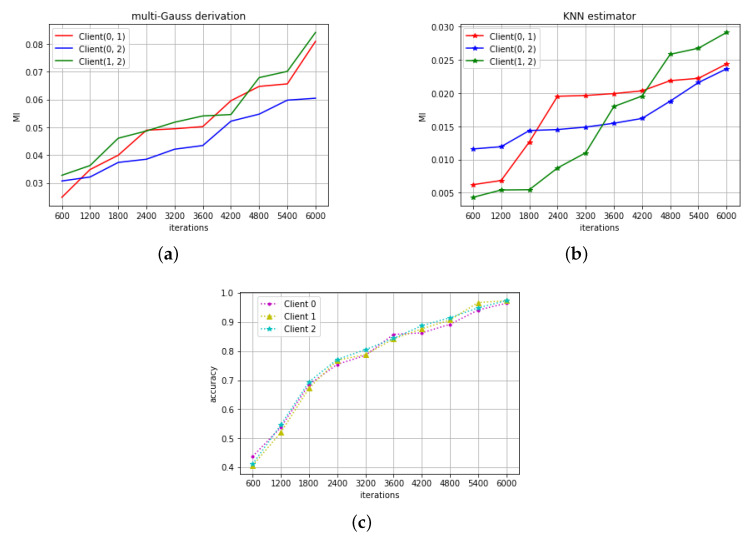
(**a**) The MI in the digit recognition task calculated by the first method (Section 3.1). (**b**) The MI in the digit recognition task calculated by the second method (Section 3.2). (**c**) The accuracy variation with the training iterations.

**Figure 3 entropy-22-00314-f003:**
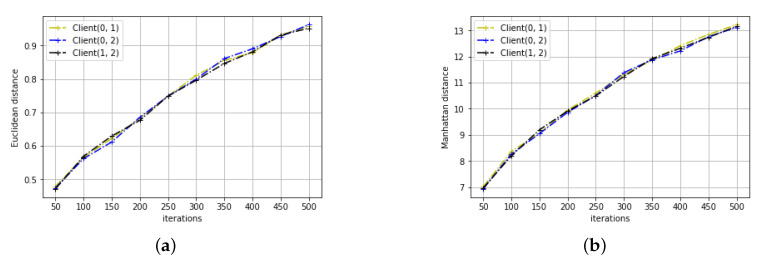
The distance metrics in the signal variation prediction task (**a**) Euclidean distance. (**b**) Manhattan distance. (**c**) Chebyshev distance.

**Figure 4 entropy-22-00314-f004:**
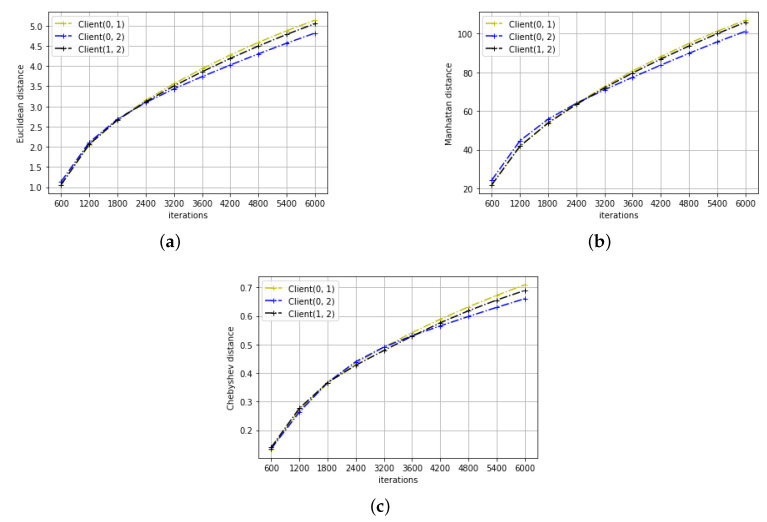
The distance metrics in the digit recognition task (**a**) Euclidean distance. (**b**) Manhattan distance. (**c**) Chebyshev distance.

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
