# Peer review of "Averaging Is Probably Not the Optimum Way of Aggregating Parameters in Federated Learning"

_entropy, 2020, doi:10.3390/e22030314_

Round 1
Reviewer 1 Report
The revised version of the manuscript is adequately improved according to suggestions including all the information and clarifications needed for publication.
Although I would like more, that authors considered utilizing centared kernal alignment (CKA) similarity measure, to compare the similarity between a weight-wise averaged representation obtained from 3 client's representation's in the selected experimental settings (not just inter-client comparison) to show the full aspects of the presented results, which would be more convincing demonstration of conclusions.
Reviewer 2 Report
The authors have already addressed my previous comments. I am fine with it.
This manuscript is a resubmission of an earlier submission. The following is a list of the peer review reports and author responses from that submission.
Round 1
Reviewer 1 Report
In the paper, the authors explore algorithmic transparency of Federated Averaging (FedAvg), a classical method for global model aggregation in (synchronous) federated learning setting by analyzing mutual information (and distances) between parameters of local models. The experimental evaluation is conducted on two learning tasks with two corresponding DNN models (RNN – time-series variation prediction & CNN – digit classification) on three peers (clients).
In general, statistical challenges of federated learning based on FedAvg (where parameters of local models are averaged element-wise with weights proportional to sizes of local datasets) are well-known and have been addressed by numerous different approaches (FedProx, PFNM, ...). However, understanding the effects of FedAvg through the information-theoretic perspective by utilizing the Mutual Information (MI) is still an unexplored and interesting topic for the interdisciplinary research community.
Although, the paper is appropriately written and structured, there are some major drawbacks that should be addressed.
1) Related work
In order to improve the scientific context and related work, the authors should at least relate to some known and recent studies utilizing the visualization of DNNs within the information plane (i.e. the plane of mutual information values of any other variables with input, and desired output) or in general utilizing mutual information for a better understanding of training dynamics and internal representations of DNN models, e.g. such as:
Saxe, Y. Bansal, J. Dapello, M. Advani, A. Kolchinsky, B. Tracey, and D. Cox. On the information bottleneck theory of deep learning. 02 2018. Shwartz-Ziv, R.; Tishby, N. Opening the Black Box of Deep Neural Networks via Information. arXiv 2017, arXiv:1703.00810. Kolchinsky, Artemy, Brendan D. Tracey, and David H. Wolpert. "Nonlinear information bottleneck." Entropy21, no. 12 (2019): 1181. Noshad, Y. Zeng, and A. O. Hero. Scalable mutual information estimation using dependence graphs. In ICASSP 2019 - 2019 IEEE International Conference on Acoustics, Speech and Signal Processing (ICASSP), pp. 2962–2966, May 2019. doi: 10.1109/ICASSP.2019.8683351. Wickstrøm, Kristoffer, et al. "Information Plane Analysis of Deep Neural Networks via Matrix-Based Renyi's Entropy and Tensor Kernels." arXiv preprint arXiv:1909.11396(2019).Most importantly, the authors should explain differences in approaches and objectives in comparison to another related study for understanding the effects of FedAvg based federated learning:
Adilova, Linara, Julia Rosenzweig, and Michael Kamp. "Information-Theoretic Perspective of Federated Learning." arXiv preprint arXiv:1911.07652 (2019).
2) Experimental setup and experiments
MAJOR:
It is not reasonable to apply classical distance measures (L1, L2, L_inf) to directly compare DNN parameter values (even of two same architectures) as many permutations of the same parameters give rise to the same model. Therefore, the presented result of increasing distances is ‘obvious’. Moreover, these distances are also not appropriate for high dimensional data and it would be better to utilize dimensionally reduction techniques (such as tSNE, UMAP…) or even use them for visualizing 2d low dimensional representations.
Additionally, the authors should explain what is the reasoning (benefits/drawbacks) behind comparing learned DNNs by MI of their parameter samples obtained from multiple 1-epoch training runs (instead of e.g. comparing inner representations across networks using approaches such as SVCCA or CKA).
The authors should also elaborate on the usage of a nonparametric k-nearest-neighbor-based method by Kraskov in relation to EDGE MI estimator by Noshad et. al. (https://github.com/mrtnoshad/EDGE). It would be beneficial to also extend the experimental evaluation with EDGE in addition to the analytical formula and KSG.
MINOR:
For training setup provide explicit details for hyper-parameters of used optimizer i.e. mini-batch stochastic gradient descent (with batch size=44?, learning rate), initialization and etc… The description of the second experiment is ambiguous and does not seem proper. Authors state that they “trained whole LeNet model on MNIST datasets (on all 10 classes?, how many epochs? is data augmentation used during pertaining? how the network is initialized, is it Image-Net pre-trained?) and afterward apply transfer learning (but not explaining for which task and dataset). Transfer learning would mean that models pre-trained on ImageNet for object detection task (or e.g. on MINST for first 5 digit classification) are fine-tuned on MINST for digit classification (or e.g. on MINST for last 5 digit classification task). Probably, authors just meant fine-tuning of the model with the same task on the same dataset applying different data augmentation techniques. When applying fine-tuning on only the last fully connected layer, it is not clear is it always initialized with the same pertained weights for each local 1-epoch training run Explicitly provide the number of how many 1-epoch training runs were conducted in order to obtain DNN parameter samples for calculating MI Please (at last) elaborate why chosen experimental setting only explores iid local datasets, neglecting non-iid case, which is a more natural setting for federated learning
Reviewer 2 Report
This paper studies a very interesting problem, i.e., federated learning. In particular, the authors investigate if the average aggregation used in most federated learning papers is a good choice. The result is interesting and may motivate other researchers to pursue better aggregation algorithms.
Strength:
The problem of federated learning is new and interesting. The authors clearly state their assumption when calculating mutual information.Weakness:
There are some grammar errors in the paper. For example, "By using two methods, one is an analytical method through the MI derivative formula under multi-dimensional Gaussian distribution assumption, and another is a discretization estimator which calculates MI between arbitrary datasets with unknown distribution based on k-nearest neighbor (KNN) distances described in[25], we estimate MI at different training phases." The experiment results are not very convincing. The authors choose very small toy dataset, small networks, and only 3 participants in federated learning. Even though the experiment design is very good, we cannot draw a general conclusion from such a small experiment.Author Response
Please see the attachment.
